# Is Mamba Capable of In-Context Learning?

Riccardo Grazzi[1,*]  Julien Siems[2,*]  Simon Schrodi[2]  Thomas Brox[2]  Frank Hutter[2]

[1]Istituto Italiano di Tecnologia, Genoa, Italy.
[2]University of Freiburg, Germany.
[*]Equal contribution.

**Abstract**  State of the art foundation models such as GPT-4 perform surprisingly well at in-context learning (ICL), a variant of meta-learning concerning the learned ability to solve tasks during a neural network forward pass, exploiting contextual information provided as input to the model. This useful ability emerges as a side product of the foundation model's massive pretraining. While transformer models are currently the state of the art in ICL, this work provides empirical evidence that Mamba, a newly proposed state space model which scales better than transformers w.r.t. the input sequence length, has similar ICL capabilities. We evaluated Mamba on tasks involving simple function approximation as well as more complex natural language processing problems. Our results demonstrate that, across both categories of tasks, Mamba closely matches the performance of transformer models for ICL. Further analysis reveals that, like transformers, Mamba appears to solve ICL problems by incrementally optimizing its internal representations. Overall, our work suggests that Mamba can be an efficient alternative to transformers for ICL tasks involving long input sequences. This is an exciting finding in meta-learning and may enable generalizations of in-context learned AutoML algorithms (like TabPFN or Optformer) to long input sequences. The code to reproduce our experiments is available at [github.com/automl/is_mamba_capable_of_icl](github.com/automl/is_mamba_capable_of_icl).

## 1 Introduction

Recent advancements in large-scale neural language modeling (Brown et al., 2020) have demonstrated that Transformer models (Vaswani et al., 2017) exhibit in-context learning (ICL) capabilities: after (self-supervised) pre-training, they can infer how to perform tasks only from input examples without the need for explicit training nor fine-tuning. This ability represents a departure from established in-weights learning of traditional machine learning and has sparked considerable academic interest as a new type of meta-learning. In contrast to standard meta-learning approaches (Hospedales et al., 2021), ICL emerges in transformer models from pre-training: without explicit training on a distribution of tasks, without bi-level optimization, and without any specific inductive bias. Recent studies advanced the understanding of how transformers can implement and learn in-context gradient-based methods when trained on distributions of simple supervised learning tasks, e.g., on linear regression tasks (Von Oswald et al., 2023; Ahn et al., 2023; Bai et al., 2023). Despite such results, whether pre-trained transformers perform in-context gradient methods on more complex tasks remains an ongoing discussion (Shen et al., 2023).

Orthogonal to these investigations into the transformer architecture, recent work proposed deep state space models to overcome limitations of transformers in processing long sequences (Tay et al., 2021), such as S4 (Gu et al., 2021a) or H3 (Gu et al., 2021b). These models merge elements from recurrent and convolutional networks with state space approaches (Kalman, 1960). However, their success on NLP tasks was limited due to problems handling dense information tasks. A key feature of state space models is that they can run the forward pass in two modes with different complexity w.r.t. the input sequence length: a parallel mode, ideal for training but with superlinear time complexity and a recurrent mode with linear time complexity and more suited for inference.

In contrast, the forward pass of transformer models has quadratic time complexity and hence it is less efficient.

This work conducts an investigation into the ICL capabilities of the recently proposed Mamba architecture (Gu and Dao, 2023), a successor to S4 and H3. Mamba has already shown its potential in different applications beyond NLP, such as visual representation learning (Zhu et al., 2024) or image segmentation (Ma et al., 2024). Concurrent to our work, the ICL capabilities of Mamba were investigated on synthetic language learning tasks by Akyürek et al. (2024) and on other tasks, including simple function classes, by Park et al. (2024). Differently from those works, we study the performance of the pre-trained Mamba model on NLP tasks, conduct a probing analysis on the representations at intermediate layers to better understand Mamba's ICL solution mechanisms, and measuring the models' ability to extrapolate beyond the context length used for training on simple function classes.

Models capable of ICL have shown good AutoML performance. A remarkable example are Prior-data Fitted Networks (PFNs) (Müller et al., 2022), which are (usually transformer) models pretrained on a range of synthetic tasks generated from a prior distribution, to approximate the posterior predictive distribution. Interestingly, by harnessing ICL and with a rich enough prior, PFNs, specifically TabPFN (Hollmann et al., 2023), exhibit strong generalization performance on real tabular data, enabling predictions on new data without the need for explicit specification and optimization of a table-specific model. Other applications of ICL relevant to AutoML are in Black-Box Optimization (Müller et al., 2023; Yang et al., 2023a; Chen et al., 2022), Learning-Curve Extrapolation (Adriaensen et al., 2024), and Time-Series Forecasting (Dooley et al., 2024). Our experimental setup for simple function classes, which closely follows the one in (Garg et al., 2022), is similar to that of PFNs, but the priors we use concern only simple regression tasks and the model only outputs the mean of the posterior predictive distribution.

**Contributions**. After introducing state-space models and Mamba more formally (Section 2), we make the following contributions:

- We demonstrate that Mamba is capable of ICL and performs on-par with transformers on simple function classes (Section 3.1) and more complex NLP tasks (Section 4). This highlights Mamba as an efficient alternative to transformers for ICL tasks entailing long sequences. In addition, we find that Mamba outperforms its predecessor S4, and RWKV (Peng et al., 2023), a recent parallel/recurrent architecture.

- Using a simple probing approach, we provide preliminary insights into the mechanism by which Mamba incrementally solves ICL tasks (Section 3.2). We find that the optimization processes exhibited by Mamba are similar to those of transformer models.

## 2  State Space Models and the Mamba Architecture

State Space Models (SSMs) are sequence-to-sequence models (when discretized) with learnable parameters which are inspired by a continuous-time model. SSMs map an input sequence $(x_1, \ldots, x_L)$ to the output sequence $(y_1, \ldots, y_L)$ by computing latent states $(h_1, \ldots, h_L)$. The map of linear time invariant SSMs such as S4 (Gu et al., 2021a) or H3 (Fu et al., 2023) can be equivalently written as

$$
\begin{aligned}
h_t &= \bar{A} h_{t-1} + \bar{B} x_t \\
y_t &= C h_t
\end{aligned}
\tag{1}
$$

$$
\begin{aligned}
K_i &= C \bar{A}^i \bar{B} \\
y_t &= \sum_{i=0}^{t-1} K_i x_{t-i}
\end{aligned}
\tag{2}
$$

where $h_0 = 0$, $\bar{A} = f_1(A, B, \Delta)$, $\bar{B} = f_2(A, B, \Delta)$, $f_1, f_2$ specify the type of discretization and $(A, B, C, \Delta)$ are learnable parameters. In particular, if $x_i \in \mathbb{R}^D$ and $h_i \in \mathbb{R}^{ND}$, then $A, B, C$ are matrices of dimensions $ND \times ND$, $ND \times D$ and $D \times ND$ respectively, while $\Delta$ is a scalar discretization parameter. To have a sufficiently large memory of the past, the dimension of $h_t$ is usually much larger than

that of $x_t$. The outputs and states in equation 1 can be computed recurrently with $O(L)$ time complexity and $O(1)$ space complexity. However, since the state update is linear in $h_{t-1}$ and $x_t$ and does not change with $t$ (linear time invariant) the outputs can also be obtained (see equation 2) by first computing $K_i$ in parallel for $i = 0, \ldots, L-1$ and then computing $(y_1, \ldots, y_L)$ through a convolution implemented via Fast Fourier Transform (FFT), which can be easily parallelized and has time complexity of $O(L \log(L))$. The recurrent mode is ideal for auto-regressive inference, since in that case parallelization is not possible, while for training the convolutional mode can fully take advantage of the parallelism of modern specialized hardware.

A fundamental limit of linear time invariant SSMs is that they do not have a mechanism to select which information to retain in the latent state based on the input sequence, and this makes them perform badly in tasks involving content-aware reasoning like selective copying, in which transformer models excel. The Mamba architecture (Gu and Dao, 2023) is a linear time varying SSM which overcomes this limit via a selection mechanism allowing the latent state dynamics to change with the current input, thus enabling selective retention of information. In particular, state and outputs of Mamba follow the recursion (starting from $h_0 = 0$)

$$h_t = \bar{A}_t h_{t-1} + \bar{B}_t x_t$$
$$y_t = C_t h_t$$

where $\bar{A}_t = \exp(A\Delta_t), \bar{B}_t = \exp(A\Delta_t)^{-1} \exp(\Delta_t A - I)\Delta_t B_t$ (zero order hold discretization), $B_t = W_1 x_t + b_1, C_t = W_2 x_t + b_2, \Delta_t = \text{softplus}(W_3 x_t + b_3)$ and $(A, W_1, W_2, W_3, b_1, b_2, b_3)$ are learnable parameters and $A$ is diagonal. The now time-varying discretization parameter $\Delta_t \in \mathbb{R}$ enables a gating mechanism which can selectively ignore the current input or reset the state. However, the selection mechanism hinders the use of convolutions for fast and parallel training. Despite this, Mamba has a similar training time as linear time invariant SSMs thanks to a hardware-aware algorithm that uses a parallel scan ($O(L)$ time complexity with only $O(\log(L))$ sequential steps) in place of FFT and stores and updates the large latent states only in the fast SRAM, rather than in GPU HBM memory. We refer to Gu and Dao (2023) for further details on Mamba's architecture.

## 3 Investigation of Simple Function Classes

In this section, we assess Mamba's ability to learn task distributions involving simple function classes. We followed the experimental protocol of Garg et al. (2022): each model is trained on a task distribution and then tested on the same distribution. This process was repeated for 4 regression task distributions, each falling into a specific function class: linear functions, sparse linear functions, 2-layer ReLU neural networks, and Decision trees. Models trained on linear functions were also tested on out-of-distribution (OOD) tasks. Differently from Garg et al. (2022), we also tested the models on tasks with more input examples than seen during training, to measure whether they can extrapolate to longer inputs. Details on each task distribution are provided in Appendix A.1.3.

We compare Mamba to a causal transformer model using the GPT2 architecture (Radford et al., 2019) and to some baselines specific to each function class (see Appendix A.1.4). We also compare with S4, a linear time invariant model, to measure the impact of the selection mechanism of Mamba. To ensure a fair comparison, the architectures of Mamba and S4 are adjusted to have a comparable number of parameters to the transformer (9.5 M). For further training details, we refer to Appendix A.1.1.

We removed the positional encoding used in the causal transformer by Garg et al. (2022), since we observed that it hinders the transformer's ability to extrapolate to longer inputs, as also shown by Press et al. (2021). Similar to Müller et al. (2022), we argue this to be more natural in the present setup as the input is actually a *set* of data points, not a sequence.

To sample the inputs of each training task for linear regression, Garg et al. (2022) used a normal distribution, while we used a skewed Gaussian distribution. Our findings show that this

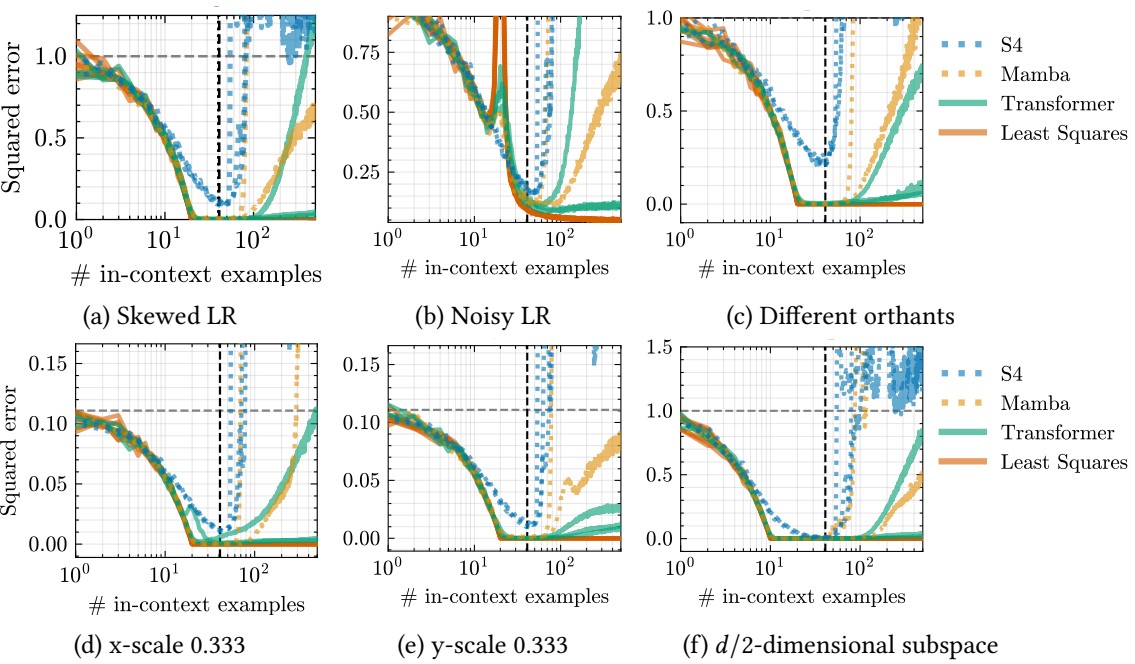

Figure 1: Comparative visualization of Mamba, S4, and transformer models (3 training seeds per method) trained on skewed linear regression and evaluated in-distribution (a) and out-of-distribution (b) - (f). The dashed vertical line indicates the number of in-context examples used for training (40).

improves robustness w.r.t. OOD tasks of both Mamba and transformers. Formally, we trained on linear functions in the set $F = \{f : f(x) = w^T x, w \in \mathbb{R}^d\}$ with input dimension $d = 20$. For each training task, we sampled $w$ from an isotropic Gaussian $\mathcal{N}(0, I_d)$ and $x_1, \ldots, x_k, x_{k+1}$ from $\mathcal{N}(0, \Sigma)$, where $\Sigma$ is a skewed covariance matrix with its eigenbasis chosen at random and the $i$th eigenvalue proportional to $1/i^2$. Following this, we set $y_i = w^T x_i$ and assembled the input prompt $P = (x_1, y_1, x_2, y_2, ..., x_k, y_k, x_{k+1})$. We used the mean squared error (MSE) $\frac{1}{k+1} \sum_{i=1}^{k+1} (\hat{y}_i - y_i)^2$ as loss function, where $\hat{y}_i$ is the output of the model corresponding to $x_i$. For all results in this section, we report the MSE/$d$ ($d = 20$), which is what is reported by Garg et al. (2022) for linear regression.

### 3.1 Analysis of in-distribution and out-of-distribution performance

Results are shown in Figures 1 and 2. In skewed linear regression, both Mamba and the Transformer model closely match the least squares baseline both in-distribution and out-of-distribution as long as the context length is shorter than the one used for training. Increasing the context length generally decreases performance, by an amount depending on the training run: two runs of the transfomer model present very little degradation while Mamba degrades substantially for all three runs. In contrast, S4 performs much worse than the least squares baseline in all setups. We hypothesize that S4's poor ICL performance is due to its linear time *invariance*. A similar hypothesis was also drawn by Gu and Dao (2023) for the task of selective copying. Similar results for Mamba and the Transformer also hold for sparse linear regression, while for neural networks and Decision trees, Mamba and the transformers are comparable and even show a promising input length extrapolation: the performance improves after the dotted vertical line indicating the number of in-context examples used during training. Interestingly, in the case of Decision trees, two out of three runs of the Transformer perform substantially better than Mamba, while Mamba's error degrades less than the one of the Transformer for ReLU neural networks. We also note that when trained on non-skewed linear regression tasks, Mamba and Transformers are less robust to OOD

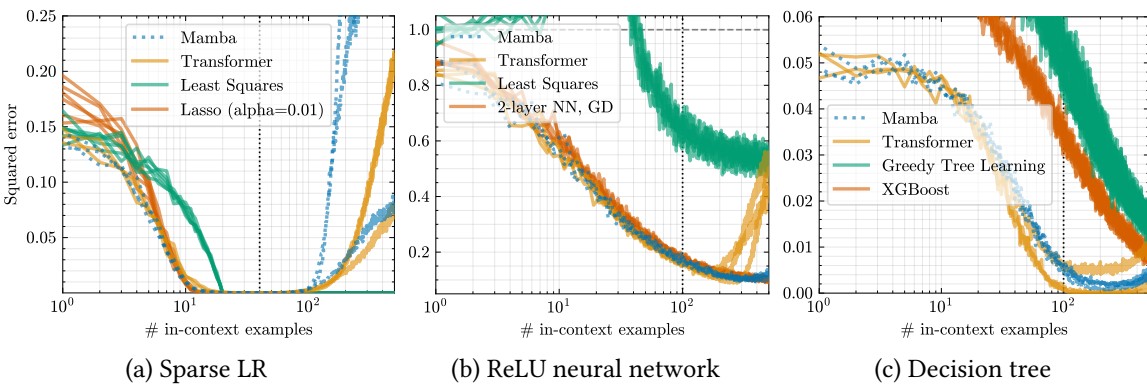

(a) Sparse LR          (b) ReLU neural network          (c) Decision tree

Figure 2: Comparison of Mamba and transformer models (3 training seeds per method) trained and tested on the same task distribution. The dotted vertical line indicates the number of in-context examples used for training (40 for sparse linear regression, 100 for ReLU neural network and Decision tree).

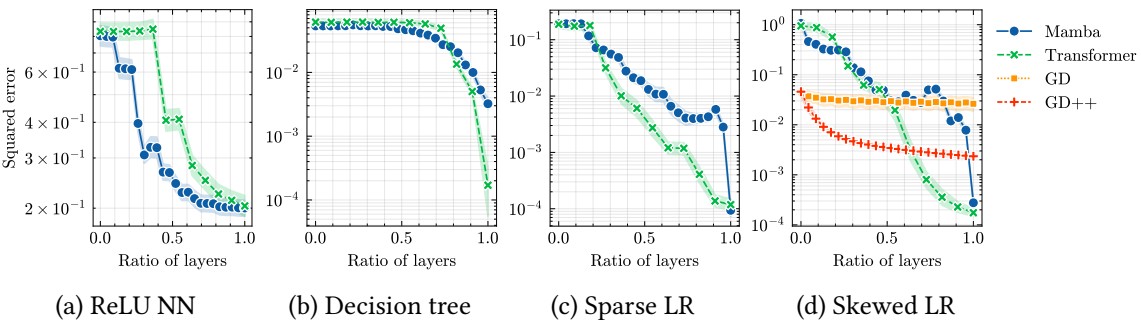

(a) ReLU NN      (b) Decision tree      (c) Sparse LR      (d) Skewed LR

Figure 3: ICL 'Learning curves' depicting the iterative optimization performance of Mamba and Transformer models on three regression tasks. Ratio of layers is the layer index divided by the total number of layers for each model (12 for the Transformer and 24 for Mamba). ReLU NN and Decision tree evaluated at $k = 100$ in-context examples and sparse/skewed LR at $k = 40$ in-context examples (i.e. the same number of in-context examples used during training).

tasks (Figure 9), even if they both perform well in-distribution (Figure 8). In the appendix we also report the performance when increasing the number of in-context examples during training for skewed linear regression (Figure 10).

### 3.2 Mechanistic understanding via probing

To better understand how the Mamba and transformer models perform ICL, we aim to test if they both employ a solution strategy akin to iterative optimization, i.e., we study if they incrementally improve their solutions layer after layer (Von Oswald et al., 2023; Ahn et al., 2023; Bai et al., 2023).

    We adopted a probing strategy similar to the one by Geva et al. (2021) for transformer language models. Differently from other probing strategies like the one by Akyürek et al. (2023), who learn a (non-linear) probe on the high-dimensional intermediate representations, this strategy learns a linear probe on the output of the decoder applied to the intermediate representations after each layer. We argue this to be a less biased probing strategy, since in our setup it reduces the degrees of freedom of the probe to just 2 parameters per task (scale and shift), since the models have scalar output.

    More specifically, our probing strategy works as follows. Let $\{(x_i, y_i)\}_i$ be i.i.d. samples belonging to one task. To compute and evaluate intermediate predictions, we use $k$, $m$ and $n$

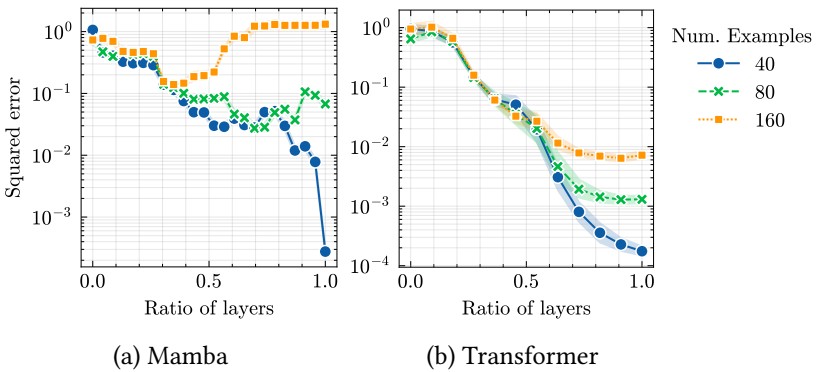

(a) Mamba        (b) Transformer

Figure 4: ICL 'Learning curves' depicting the iterative optimization performance of Mamba and Transformer models (both trained with 40 in-context examples) on skewed linear regression when tested on varying the number of in-context examples from 40 to 160.

examples for the train, validation, and test set, respectively. First, we separately feed the prompts $(x_1, y_1, x_2, y_2, ..., x_k, y_k, x_j)$ for $j = k+1, ..., k+m+n$ to the model, to obtain for each token $x_i$ the internal representations $z_{l,i}$ for each layer $l$ and the final prediction $\hat{y}_i = g(z_{L,i})$, where $g$ is the decoder and $L$ is the index of the last layer. Then for each layer $l$ and each token $i > k$, we first obtain intermediate scalar predictions $\tilde{y}_{l,i} = g(z_{l,i})$. Since $g$ is not meant to be used on intermediate representations, we finally adjust the predictions by computing $\hat{y}_{l,i} = a_l g(z_{l,i}) + b_l$, where the scalar scale and shift parameters $a_l$ and $b_l$ are obtained by least squares on $(\tilde{y}_{l,k+1}, y_{k+1}) \ldots (\tilde{y}_{l,k+m}, y_{k+m})$ (i.e., using the validation tokens). We can then measure the accuracy of these intermediate predictions with the normalized mean squared error on the test tokens $(nd)^{-1} \sum_{j=1}^{n} (\hat{y}_{l,k+m+j} - y_{k+m+j})^2$ ($d = 20$). We ran our analysis over 128 tasks (e.g. different linear regression weight vectors) with $k$ as specified in Figures 3 and 4, $m$=10 and $n$=40. We conducted the probing analysis on Mamba and transformer models trained on skewed linear regression, sparse linear regression, ReLU neural networks, and Decision trees.

For skewed linear regression, we additionally compared to Gradient Descent (GD) and GD$^{++}$ as done in Von Oswald et al. (2023). GD$^{++}$ is a version of preconditioned gradient descent in which the data samples undergo a transformation $(x_i \leftarrow (I - \gamma XX^T)x_i)$ and we tuned $\gamma$ for optimal performance via a grid search (c.f., Figure 11 in Appendix), while the step-size was set for each task to the theoretically optimal value, i.e., as $2/(L + \mu)$, where $L, \mu$ are the largest and smallest eigenvalues of the empirical covariance matrix of the task. We ran GD and GD$^{++}$ for 24 iterations to match the 24 layers of our Mamba model. Figure 3 provides strong evidence to the hypothesis that both Mamba and the transformer employ an iterative solution scheme on the skewed and sparse linear regression task (Figure 3d), since the log-MSE decreases (almost) linearly. In ReLU neural networks, the error also decreases somewhat gradually after some layers. Interestingly, for Decision trees the error stays high for both models for more than half of the initial layers. The comparison between GD, GD$^{++}$, Mamba, and the transformer in Figure 3d reveals that both GD and GD$^{++}$ are outperformed by Mamba and the Transformer, while GD converges more slowly due to the tasks having skewed covariance.

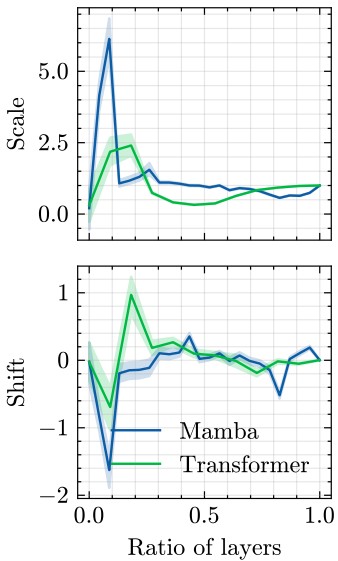

Figure 5: Skewed LR

In Figure 4 we show how, similarly to Figure 1a, having more in-context examples than the ones used during training negatively affects the learning curves for linear regression. Both models never seem to take advantage of the additional number of examples in this task: the performance with 40 in context examples (the number used for training) is the best (or close) for all layers. Moreover, we note that the learning curve for Mamba with 160 in-context examples exhibits a clear U-shape: the model actually found a good solution in intermediate layers but ultimately degrades.

In Figure 5, we compare the scale and shift parameters estimated by the linear probing model for Mamba and the transformer in the context of Skewed Linear Regression. Additionally, for sparse linear regression, ReLU neural networks, and Decision trees, we present the findings in Figure 12 in the Appendix. Interestingly, for linear regression and ReLU neural networks, the linear probing model applies only minor modifications, with scale values near one and shift values approaching zero after the first layers. We also observe that for all tasks, the variance across tasks of the estimated scale and shift is higher in the early layers of the model and quickly approaches zero as we move towards the output layer.

## 4 Investigation of Simple NLP Tasks

In this section, we evaluated the ICL performance of various pre-trained Mamba language models, with parameter counts from 130 million to 2.8 billion. The pre-training for all variants was done on the Pile dataset (Gao et al., 2020), while the Mamba 2.8B (SP) checkpoint underwent further fine-tuning on ca. 600 billion tokens from the SlimPajama dataset.

We compared the Mamba variants to another RNN model with linear state dynamics named RWKV (Peng et al., 2023), which was also pre-trained on the Pile[1], and popular transformer-based language models, such as LLama (Touvron et al., 2023), Pythia (Biderman et al., 2023), and GPT-J 6B (Wang and Komatsuzaki, 2021). We did not compare to S4 because we are not aware of S4 models pretrained on the Pile.

We followed the experimental protocol of Hendel et al. (2023), which tested 27 NLP tasks spanning a wide range of categories, including algorithmic tasks (e.g., list element extraction), translation (e.g., English to Spanish), linguistic tasks (e.g., singular to plural conversion), and knowledge-based tasks (e.g., identifying country-capital pairs). For evaluation, we used the same datasets as Hendel et al. (2023) except for the algorithmic tasks, which were randomly generated (we use the same generation parameters). We report the mean accuracy over 400 generated test sets per task, each having five in-context examples.

Figure 6 shows that the ICL performance improves for all models with increasing number of parameters. Notably, Mamba 2.8B achieves an ICL accuracy close to LLama 7B, and on par with GPT-J and Pythia models. In addition, we find that Mamba consistently outperforms the similarly scalable RWKV at comparable parameter sizes. We provide a detailed table of per-task accuracies in Appendix A.2.

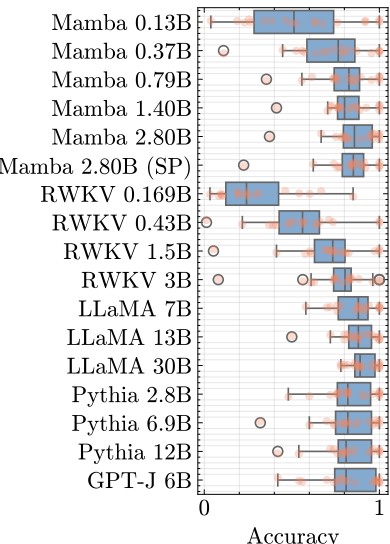

Figure 6: Accuracy of Mamba variants compared to RWKV and other transformer-based models on in-context NLP tasks.

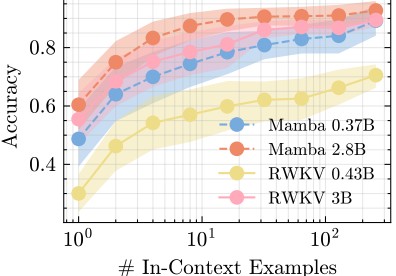

Figure 7: Average task accuracy for increasing context length. Reported are the 5/95% confidence intervals.

---

[1] We used RWKV v4 checkpoints trained on the Pile dataset by the authors available on huggingface.

Finally, we find that Mamba scales well with the number of in-context examples; see Figure 7. Particularly, Mamba 0.37B and 2.8B maintain a considerable performance edge over RWKV 0.43B and 3B, respectively.

## 5 Related Work

**In-context learning capabilities in transformer-based models**. Brown et al. (2020) introduced GPT-3 and were the first to show the remarkable in-context learning (ICL) abilities of large language models. They showed that larger models tend to utilize more in-context information compared to smaller models. Other work by Garg et al. (2022) showed that even small transformers exhibit ICL abilities on linear modeling tasks. These small transformer-based models have been subject to in-depth analysis by recent work (Dai et al., 2022; Von Oswald et al., 2023; Ahn et al., 2023; Bai et al., 2023) that showed that they can implement mechanisms akin to gradient descent or more complex optimization algorithms in the forward pass.

On a different tangent, Olsson et al. (2022) found that a two-layer attention-only network can develop the so-called "induction heads" mechanism, which outputs the token succeeding a previous instance of the current token, precisely when its ICL performance increases. Chan et al. (2022) investigated properties of the data-distribution which lead to the emergence of ICL abilities, while Reddy (2024) identified factors for the abrupt emergence of the induction heads. Another line of work (Hendel et al., 2023; Todd et al., 2024) showed that some intermediate output of the forward pass of transformer models, named "task vectors", encodes most of the information for the in-context task. In particular, merging a task vector with another from a different in-context task allows the model to solve a combination of the two tasks.

**In-context learning benchmarks**. Garg et al. (2022) tested transformers abilities to learn task distributions from simple function classes, like linear regression or Decision trees. Hendel et al. (2023) tested pre-trained LLMs on a suite of NLP tasks such as English to French translation. Lastly, Akyürek et al. (2024) proposed to test NLP models' ability to learn formal languages.

**Sub-quadratic transformer alternatives**. Transformer-based models have a forward pass with time complexity scaling quadratically with respect to the input sequence length. To reduce the amount of computation, previous work proposed to introduce sparsity into the attention layers (Child et al., 2019; Qiu et al., 2019; Beltagy et al., 2020), where the tokens only attend a subset of the other tokens. Other works by Wang et al. (2020) proposed a low-rank factorization of the attention mechanism. Others proposed linear attention, where the attention weights are computed using possibly normalized dot-products without the softmax (Katharopoulos et al., 2020; Choromanski et al., 2020; Kasai et al., 2021; Peng et al., 2021; Yang et al., 2023b). Intriguingly, causal linear attention can be re-formulated either as a linear RNN or as the ratio between two linear RNNs, and hence its time complexity scales linearly with the length of the input sequence.

Recently, state-space sequence models gained significant interest due their strong performance. They are broadly related to recurrent neural networks with ideas from classical state space models, such as the Kalman filter (Kalman, 1960), particle filters (Gordon et al., 1993), or hidden Markov models (Baum et al., 1970; Rabiner, 1989). There exist various state-space models: S4 (Gu et al., 2021a), DSS (Gupta et al., 2022), S5 (Smith et al., 2023), GSS (Mehta et al., 2023), H3 (Fu et al., 2023), Selective S4 (Wang et al., 2023), or RWKV (Peng et al., 2023). While they have been successfully applied to audio and vision (Goel et al., 2022; Nguyen et al., 2022), they lacked performance for text. The recently proposed Mamba (Gu and Dao, 2023) exhibits great performance in large language modeling and similar scaling properties as state of the art transformer models.

## 6 Broader Impact Statement

In-context learning (ICL) is an emergent phenomenon that drives the generalization skills of large language models (LLMs). While the widespread use of a poorly-understood technology like ICL

may have many potential negative societal impacts, our work contributes to mitigate this problem by helping to build a mechanistic understanding of ICL. This understanding is crucial to build better, more explainable and more trustworthy AI systems in the future. Since the favourable scaling of Mamba (Gu and Dao, 2023) makes it a strong candidate for dealing with long context sizes, we do believe it to be particularly important to understand the mechanisms behind ICL in this architecture, and we thus focus on this aspect in our paper.

## 7 Limitations

Our investigation into the Mamba architecture's in-context learning (ICL) capabilities shows promise as an efficient alternative to transformer models for processing long input sequences, yet it is not without limitations. The study's focus on simple function approximation and natural language processing (NLP) tasks raises questions about the generalizability of our findings to other domains like image or audio analysis, suggesting the need for broader research. The linear probing method used to understand Mamba's approach may oversimplify the model's complex optimization processes, indicating that more sophisticated probing techniques could yield deeper insights. Additionally, our evaluation covered a limited range of model sizes and configurations, and a more comprehensive comparison is necessary to fully assess Mamba's performance relative to extensively optimized transformer models. The scalability of Mamba with increasing in-context examples and its computational efficiency also remain underexplored, highlighting areas for future investigation to better understand its practical applicability and potential across a wider range of tasks and settings.

## 8 Conclusions and Future Directions

In this work, we have demonstrated that the recently proposed Mamba architecture is capable of effective in-context-learning (ICL) across tasks involving simple function approximation as well as more complex natural language processing problems. Our analysis showed that Mamba performs on par with transformer models, while also outperforming the S4 and RWKV baselines. We provide initial evidence that Mamba appears to solve ICL problems by incrementally refining its internal representations in a manner akin to an iterative optimization strategy, as transformers do. Overall, our findings suggest that Mamba can be an efficient and performant alternative to transformers for ICL involving longer input sequences.

In future work, it would be interesting to replace the attention-based backbone of existing in-context learned AutoML algorithms, such as TabPFN (Hollmann et al., 2023), ForecastPFN (Dooley et al., 2024) or Optformer (Chen et al., 2022), enabling them to handle longer sequences.

**Acknowledgements**. We would like to thank Alma Lindborg, André Biedenkapp and Samuel Müller (alphabetic order) for their constructive feedback in preparation of this paper.

This work was supported in part by the European Union – NextGenerationEUPNRR MUR, EU project 101070617 entitled "European Lighthouse on Secure and Safe AI", and EU project 101120237 entitled "European Lighthouse of AI for Sustainability"

This work was supported in part by the Deutsche Forschungsgemeinschaft (DFG, German Research Foundation) under grant number 417962828, and the Bundesministerium für Umwelt, Naturschutz, nukleare Sicherheit und Verbraucherschutz (BMUV, German Federal Ministry for the Environment, Nature Conservation, Nuclear Safety and Consumer Protection) based on a resolution of the German Bundestag (67KI2029A)

The authors acknowledge support by the state of Baden-Württemberg through bwHPC and the German Research Foundation (DFG) through grant no INST 39/963-1 FUGG. We acknowledge funding by the European Union (via ERC Consolidator Grant DeepLearning 2.0, grant no. 101045765). Views and opinions expressed are however those of the author(s) only and do not necessarily reflect

those of the European Union or the European Research Council. Neither the European Union nor the granting authority can be held responsible for them.

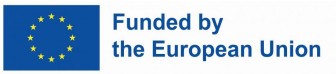

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

## A  Appendix

### A.1  ICL for simple function classes

We follow the experimental setup of Garg et al. (2022) by building on their MIT-Licensed code at `https://github.com/dtsip/in-context-learning`.

**A.1.1  Model parameters**. As in Garg et al. (2022), we used a GPT-2 (Radford et al., 2019) model with embedding size 256, 12 layers and 8 heads, resulting in 9.5 million parameters. As mentioned in the main text, we removed the positional encoding to improve the input length extrapolation. Mamba's $N$ parameter is set to its default value 16, while we set Mamba's $D$ parameter to 256, matching the transformer's embedding dimension, while we set the number of Mamba's layers to 24, doubling the ones of the transformer (12). This is done because each Mamba block can be roughly seen as the fusion of a MLP and SSM, and has roughly half the number of parameters of a transformer block, which can be divided into an MLP and an attention part. For a fair comparison, S4 also uses 24 layers, however we set the embedding size to 435 to match the 10 million parameters of Mamba and the transformer. We note that a similar comparison between S4 and transformer models, also testing for input length extrapolation, was done by Lee et al. (2023). However, we used models with substantially more parameters (10 millions vs. 500 thousands) and transformers without positional encoders.

**A.1.2  Training details**. We adopted the same experimental setup used for the transformer model by (Garg et al., 2022) for Mamba, the transformer, and S4. At each training step, we computed the loss on a mini-batch of 64 prompts, each corresponding to a task sampled from a task distribution. We used no dropout in our experiments. We adopt a curriculum learning strategy: starting with training points of the lower-dimensional subspace and fewer input examples per prompt, and increasing the dimensionality of the subspace and the number of in-context examples every 2000 steps. For more details on the training, we refer to (Garg et al., 2022, A.2 Training).

Differently for Garg et al. (2022), we use cosine annealing (from PyTorch) for Mamba, Transformer and S4, as we observed that it consistently improved performance.

The training was done Nvidia RTX 2080 GPUs and each training had a duration from 20 hours (for skewed/linear/sparse regression) up to 24 hours (for ReLU neural network and Decision tree). The evaluation for up to 500 ICL examples was done an Nvidia A100 GPU and took 16 hours for skewed/linear/sparse regression and up to 24 hours for ReLU neural network and Decision tree.

**A.1.3  Simple task distributions**. Below, we describe how tasks are sampled for each task distribution that we considered. Task distributions are the same ones as in Garg et al. (2022).

For each task we first sampled $x_1, \ldots, x_k, x_{k+1}$ input points Then, for each point we sampled the output $y_i$ as a function of $x_i \in \mathbb{R}^d$, possibly adding noise. Finally, the prompt $P = x_1, y_1, \ldots, x_k, y_k, x_{k+1}$ was passed as input to the model, which can be divided in context $x_1, y_1), \ldots, x_k, y_k$, and query point $x_{k+1}$. We now describe how each input-output pair $(x_i, y_i)$ is computed for each task for different task distributions. We set the number of input dimensions $d = 20$ for all task distributions.

**Linear regression**. First sample $w \sim \mathcal{N}(0, I_d)$, then for $i = 1, \ldots, k+1$ sample $x_i \sim \mathcal{N}(0, I)$ and $y_i = w^T x_i$.

**Skewed linear regression (Skewed LR)**. First sample $w \sim \mathcal{N}(0, I_d)$. Then sample a $d \times d$ matrix $A$ from a normal distribution, compute the SVD $A = USV$ and the tranformation $B = U \text{diag}((1, 1/4, \ldots, 1/d^2))U^\top$. Finally for $i = 1, \ldots, k+1$ sample $\hat{x}_i \sim \mathcal{N}(0, I_d)$ and compute $x_i = B\hat{x}_i$ and $y_i = w^T x_i$.

**Sparse linear regression**. First sample $w \sim \mathcal{N}(0, I_d)$, then set all coordinates of $w$ except $s = 3$ to zero. Finally, for $i = 1, \ldots, k+1$ sample $x_i \sim \mathcal{N}(0, I)$ and $y_i = w^T x_i$.

**Noisy linear regression (Noisy LR).** First sample $w \sim \mathcal{N}(0, I)$, then for $i = 1, \ldots, k+1$ sample $x_i \sim \mathcal{N}(0, I)$ and $y_i = w^T x_i + \varepsilon_i$, with scalar noise $\varepsilon_i \sim \mathcal{N}(0, 1)$.

**Different orthants.** As linear regression, but the sign of each coordinate is randomly sampled, where every in-context example lies in one quadrant, while the query input lies in another with high probability.

**$d/2$-dimensional subspace.** As linear regression, but the coordinates from $\lceil d/2 \rceil$ to $d$ are set to 0 for each $x_i$.

**ReLU neural network.** These networks represents functions of the form $f(x) = \sum_{i=1}^{r} \alpha_i \sigma(w_i^T x)$, where $\alpha_i \in \mathbb{R}$, $w_i \in \mathbb{R}^d$ and $\sigma(\cdot) = \max(0, \cdot)$ is the ReLU activation function. To generate a random prompt $P = (x_1, f(x_1), ..., x_k, f(x_k), x_{k+1})$, we sample prompt inputs $x_i$'s from $N(0, I_d)$, along with network parameters $\alpha_i$'s and $w_i$'s from $N(0, 2/r)$ and $N(0, I_d)$ respectively. We set the input dimension $d$ to 20 and the number of the hidden neurons $r$ to 100.

**Decision tree.** We consider the class of depth 4 Decision trees with 20 dimensional inputs. A function $f$ in this class is represented by a full binary tree (with 16 leaf nodes). Each non-leaf node is associated with a coordinate of the input, and each leaf node is associated with a target value. To evaluate $f$ on an input $x$, we traverse the tree starting from the root node. We move to the right child if the coordinate associated with the current node is positive, and move to the left child otherwise (that is, the threshold at each node is 0). The function $f(x)$ is given by the value associated with the leaf node reached at the end. To sample a random prompt $P = (x_1, f(x_1), ..., x_k, f(x_k), x_{\text{query}})$, we draw prompt inputs $x_i$'s and $x_{\text{query}}$ from $N(0, I_d)$. The function $f$ corresponds to a tree where the coordinates associated with the non-leaf nodes are drawn uniformly at random from $\{1, \ldots, d\}$ and the values associated with the leaf nodes are drawn from $N(0, 1)$.

**A.1.4 Other baselines.** As in Garg et al. (2022), we compared also with task distribution specific baselines, which we will discuss below. We refer to (Garg et al., 2022, Appendix A.3) for more details.

**Least Squares.** Fits an ordinary least squares estimator to the in-context examples.

**Lasso.** Fits a LASSO estimator to the in-context examples with a specified L1 regularization parameter.

**n-Nearest neighbor.** We average the predictions of the 3 in-context examples closest in euclidean distance to the query point $x_{k+1}$.

**Averaging.** It computes the query prediction $\hat{y}_{k+1} = \hat{w}^\top x_{k+1}$, with $\hat{w} = \frac{1}{k} \sum_{i=1}^{k} x_i y_i$.

**2-layer NN, GD..** A 2-layer NN with the same number of hidden neurons used in the task distribution, trained on the in-context examples using ADAM.

**Greedy tree learning.** It learns a Decision tree greedily using scikit-learn's Decision tree regressor (Chen and Guestrin, 2016) with default parameters and max_depth equal to 2.

**Tree boosting.** We use the XGBoost library (Chen and Guestrin, 2016) to learn an ensemble of 50 Decision trees with maximum depth 4 and learning rate 0.1.

**A.1.5 Additional results.** We provide additional results on Mamba and transformers on non-skewed linear regression in Figure 8. We show its out-of-distribution performance in Figure 9. Differently from the results in the main text, instead of cosine annealing we use the same learning rates used by Garg et al. (2022) for both Mamba and the transformer model. As pointed out in the main text, we find that both Mamba and the transformer perform worse out-of-distribution compared to when they are trained on skewed linear regression.

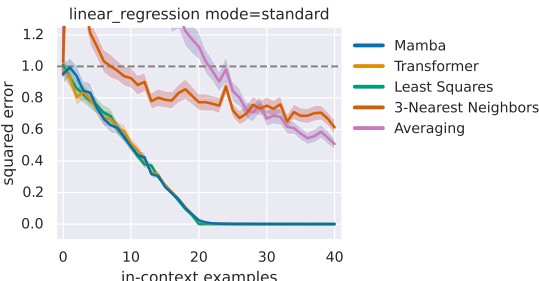

Figure 8: Mamba compares with transformers when trained and tested on the same unskewed linear regression task distribution. We report one training run for each plot and method

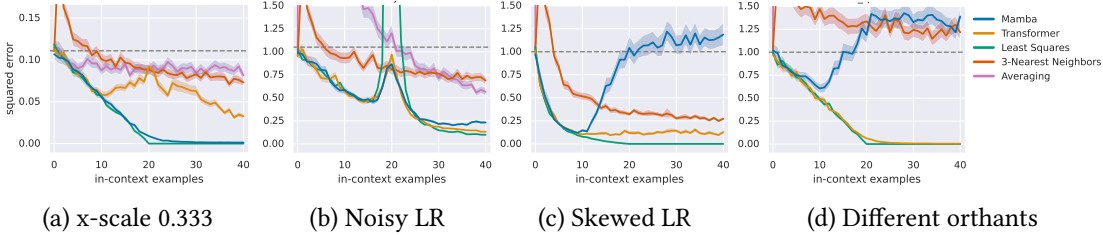

(a) x-scale 0.333   (b) Noisy LR   (c) Skewed LR   (d) Different orthants

Figure 9: OOD Distribution tests for Mamba and Transformer trained on (non-skewed) linear regression data. We report one training run for each method.

In Figure 10, we observe how the context length extrapolation performance is affected by using a higher number of examples during training for skewed linear regression tasks. In particular, the window where both models obtain good performance is wider the higher the number of in-context examples during training, with the transformer generally exhibiting a much lower degree of degradation.

**A.1.6 ICL learning curves**. The scale and shift were estimated for each layer and each task and are reported in Figure 12. We note that for skewed linear regression and ReLU neural networks the value for scale approaches 1, and the one for shift approaches 0 relatively quickly.

**A.2 ICL for NLP tasks**

We added Mamba and RKWV to the code provided by (Hendel et al., 2023) at github.com/roeehendel/icl_task_vectors. We leveraged their results for GPT-J, Llama and Pythia and provided our experimental results for Mamba and RWKV. A detailed break down of the per task performance of each model is given in Table 1.

Table 1: Complete results of the main experiment for all tasks and models.

| Model | Task type | method
Task name | Baseline | Regular |
|---|---|---|---|---|
| GPT-J 6B | Algorithmic | List first | 0.30 | 0.98 |
| | | List last | 0.24 | 1.00 |
| | | Next letter | 0.16 | 0.86 |
| | | Prev letter | 0.10 | 0.42 |
| | | To lower | 0.00 | 1.00 |
| | | To upper | 0.00 | 1.00 |
| | Knowledge | Country capital | 0.19 | 0.80 |

| Model | Task type | method
Task name | Baseline | Regular |
|---|---|---|---|---|
| | | Location continent | 0.03 | 0.70 |
| | | Location religion | 0.09 | 0.78 |
| | | Person language | 0.02 | 0.82 |
| | Linguistic | Antonyms | 0.43 | 0.78 |
| | | Plural singular | 0.08 | 0.98 |
| | | Present simple gerund | 0.00 | 0.98 |
| | | Present simple past simple | 0.02 | 0.96 |
| | Translation | En es | 0.14 | 0.56 |
| | | En fr | 0.16 | 0.54 |
| | | Es en | 0.06 | 0.74 |
| | | Fr en | 0.13 | 0.76 |
| LLaMA 13B | Algorithmic | List first | 0.77 | 1.00 |
| | | List last | 0.07 | 0.92 |
| | | Next letter | 0.31 | 0.94 |
| | | Prev letter | 0.05 | 0.50 |
| | | To lower | 0.00 | 1.00 |
| | | To upper | 0.00 | 1.00 |
| | Knowledge | Country capital | 0.17 | 0.86 |
| | | Location continent | 0.01 | 0.80 |
| | | Location religion | 0.10 | 0.84 |
| | | Person language | 0.02 | 0.88 |
| | Linguistic | Antonyms | 0.19 | 0.80 |
| | | Plural singular | 0.24 | 0.88 |
| | | Present simple gerund | 0.00 | 0.96 |
| | | Present simple past simple | 0.01 | 0.98 |
| | Translation | En es | 0.05 | 0.82 |
| | | En fr | 0.15 | 0.84 |
| | | Es en | 0.29 | 0.88 |
| | | Fr en | 0.25 | 0.72 |
| LLaMA 30B | Algorithmic | List first | 0.96 | 1.00 |
| | | List last | 0.02 | 0.96 |
| | | Next letter | 0.30 | 0.96 |
| | | Prev letter | 0.02 | 0.80 |
| | | To lower | 0.00 | 1.00 |
| | | To upper | 0.00 | 1.00 |
| | Knowledge | Country capital | 0.27 | 0.88 |
| | | Location continent | 0.01 | 0.86 |
| | | Location religion | 0.05 | 0.88 |
| | | Person language | 0.01 | 0.90 |
| | Linguistic | Antonyms | 0.37 | 0.82 |
| | | Plural singular | 0.21 | 0.90 |
| | | Present simple gerund | 0.00 | 0.98 |
| | | Present simple past simple | 0.02 | 1.00 |
| | Translation | En es | 0.07 | 0.78 |
| | | En fr | 0.10 | 0.86 |
| | | Es en | 0.24 | 0.88 |
| | | Fr en | 0.20 | 0.78 |
| LLaMA 7B | Algorithmic | List first | 0.87 | 1.00 |
| | | List last | 0.03 | 1.00 |
| | | Next letter | 0.03 | 0.88 |

| Model | Task type | method
Task name | Baseline | Regular |
|---|---|---|---|---|
| | | Prev letter | 0.04 | 0.58 |
| | | To lower | 0.00 | 1.00 |
| | | To upper | 0.00 | 1.00 |
| | Knowledge | Country capital | 0.28 | 0.86 |
| | | Location continent | 0.02 | 0.72 |
| | | Location religion | 0.12 | 0.94 |
| | | Person language | 0.02 | 0.78 |
| | Linguistic | Antonyms | 0.33 | 0.76 |
| | | Plural singular | 0.15 | 0.88 |
| | | Present simple gerund | 0.00 | 0.90 |
| | | Present simple past simple | 0.02 | 0.92 |
| | Translation | En es | 0.07 | 0.76 |
| | | En fr | 0.04 | 0.88 |
| | | Es en | 0.21 | 0.92 |
| | | Fr en | 0.15 | 0.70 |
| Mamba 0.13B | Algorithmic | List first | 0.66 | 0.92 |
| | | List last | 0.02 | 0.92 |
| | | Next letter | 0.24 | 0.76 |
| | | Prev letter | 0.06 | 0.04 |
| | | To lower | 0.00 | 1.00 |
| | | To upper | 0.00 | 1.00 |
| | Knowledge | Country capital | 0.07 | 0.19 |
| | | Location continent | 0.01 | 0.50 |
| | | Location religion | 0.02 | 0.63 |
| | | Person language | 0.01 | 0.53 |
| | Linguistic | Antonyms | 0.26 | 0.34 |
| | | Plural singular | 0.09 | 0.46 |
| | | Present simple gerund | 0.00 | 0.68 |
| | | Present simple past simple | 0.01 | 0.63 |
| | Translation | En es | 0.07 | 0.23 |
| | | En fr | 0.16 | 0.39 |
| | | Es en | 0.07 | 0.26 |
| | | Fr en | 0.10 | 0.27 |
| Mamba 0.37B | Algorithmic | List first | 0.09 | 1.00 |
| | | List last | 0.02 | 0.95 |
| | | Next letter | 0.19 | 0.87 |
| | | Prev letter | 0.07 | 0.11 |
| | | To lower | 0.00 | 1.00 |
| | | To upper | 0.00 | 1.00 |
| | Knowledge | Country capital | 0.10 | 0.57 |
| | | Location continent | 0.00 | 0.64 |
| | | Location religion | 0.06 | 0.56 |
| | | Person language | 0.01 | 0.79 |
| | Linguistic | Antonyms | 0.33 | 0.68 |
| | | Plural singular | 0.09 | 0.74 |
| | | Present simple gerund | 0.00 | 0.83 |
| | | Present simple past simple | 0.00 | 0.80 |
| | Translation | En es | 0.07 | 0.45 |
| | | En fr | 0.12 | 0.53 |
| | | Es en | 0.07 | 0.79 |

| Model | Task type | method
Task name | Baseline | Regular |
|---|---|---|---|---|
| | | Fr en | 0.08 | 0.70 |
| Mamba 0.79B | Algorithmic | List first | 0.76 | 1.00 |
| | | List last | 0.03 | 0.95 |
| | | Next letter | 0.12 | 0.85 |
| | | Prev letter | 0.03 | 0.35 |
| | | To lower | 0.00 | 1.00 |
| | | To upper | 0.00 | 1.00 |
| | Knowledge | Country capital | 0.15 | 0.76 |
| | | Location continent | 0.02 | 0.73 |
| | | Location religion | 0.12 | 0.74 |
| | | Person language | 0.00 | 0.83 |
| | Linguistic | Antonyms | 0.45 | 0.76 |
| | | Plural singular | 0.07 | 0.86 |
| | | Present simple gerund | 0.00 | 0.90 |
| | | Present simple past simple | 0.01 | 0.86 |
| | Translation | En es | 0.11 | 0.56 |
| | | En fr | 0.14 | 0.61 |
| | | Es en | 0.09 | 0.82 |
| | | Fr en | 0.20 | 0.75 |
| Mamba 1.40B | Algorithmic | List first | 0.68 | 1.00 |
| | | List last | 0.03 | 0.93 |
| | | Next letter | 0.07 | 0.77 |
| | | Prev letter | 0.05 | 0.41 |
| | | To lower | 0.00 | 1.00 |
| | | To upper | 0.00 | 1.00 |
| | Knowledge | Country capital | 0.15 | 0.76 |
| | | Location continent | 0.01 | 0.77 |
| | | Location religion | 0.06 | 0.77 |
| | | Person language | 0.00 | 0.84 |
| | Linguistic | Antonyms | 0.37 | 0.78 |
| | | Plural singular | 0.07 | 0.87 |
| | | Present simple gerund | 0.00 | 0.89 |
| | | Present simple past simple | 0.01 | 0.87 |
| | Translation | En es | 0.09 | 0.72 |
| | | En fr | 0.14 | 0.70 |
| | | Es en | 0.11 | 0.82 |
| | | Fr en | 0.14 | 0.72 |
| Mamba 2.80B | Algorithmic | List first | 0.70 | 1.00 |
| | | List last | 0.13 | 0.97 |
| | | Next letter | 0.02 | 0.95 |
| | | Prev letter | 0.04 | 0.37 |
| | | To lower | 0.00 | 1.00 |
| | | To upper | 0.00 | 1.00 |
| | Knowledge | Country capital | 0.15 | 0.80 |
| | | Location continent | 0.03 | 0.81 |
| | | Location religion | 0.12 | 0.79 |
| | | Person language | 0.02 | 0.89 |
| | Linguistic | Antonyms | 0.38 | 0.79 |
| | | Plural singular | 0.14 | 0.96 |
| | | Present simple gerund | 0.00 | 0.96 |

| Model | Task type | method Task name | Baseline | Regular |
|---|---|---|---|---|
| | | Present simple past simple | 0.01 | 0.93 |
| | Translation | En es | 0.03 | 0.75 |
| | | En fr | 0.07 | 0.67 |
| | | Es en | 0.11 | 0.83 |
| | | Fr en | 0.16 | 0.81 |
| Mamba 2.80B (SP) | Algorithmic | List first | 0.84 | 1.00 |
| | | List last | 0.07 | 0.92 |
| | | Next letter | 0.17 | 0.84 |
| | | Prev letter | 0.02 | 0.23 |
| | | To lower | 0.00 | 1.00 |
| | | To upper | 0.00 | 1.00 |
| | Knowledge | Country capital | 0.20 | 0.84 |
| | | Location continent | 0.06 | 0.86 |
| | | Location religion | 0.12 | 0.84 |
| | | Person language | 0.04 | 0.87 |
| | Linguistic | Antonyms | 0.36 | 0.77 |
| | | Plural singular | 0.11 | 0.90 |
| | | Present simple gerund | 0.00 | 0.91 |
| | | Present simple past simple | 0.01 | 0.90 |
| | Translation | En es | 0.08 | 0.74 |
| | | En fr | 0.15 | 0.62 |
| | | Es en | 0.11 | 0.83 |
| | | Fr en | 0.16 | 0.76 |
| Pythia 12B | Algorithmic | List first | 0.53 | 0.96 |
| | | List last | 0.09 | 1.00 |
| | | Next letter | 0.15 | 0.76 |
| | | Prev letter | 0.00 | 0.42 |
| | | To lower | 0.02 | 1.00 |
| | | To upper | 0.00 | 1.00 |
| | Knowledge | Country capital | 0.19 | 0.82 |
| | | Location continent | 0.01 | 0.80 |
| | | Location religion | 0.07 | 0.78 |
| | | Person language | 0.01 | 0.86 |
| | Linguistic | Antonyms | 0.34 | 0.74 |
| | | Plural singular | 0.18 | 0.84 |
| | | Present simple gerund | 0.00 | 0.96 |
| | | Present simple past simple | 0.01 | 0.94 |
| | Translation | En es | 0.10 | 0.72 |
| | | En fr | 0.16 | 0.54 |
| | | Es en | 0.05 | 0.80 |
| | | Fr en | 0.14 | 0.80 |
| Pythia 2.8B | Algorithmic | List first | 0.69 | 1.00 |
| | | List last | 0.06 | 1.00 |
| | | Next letter | 0.42 | 0.90 |
| | | Prev letter | 0.01 | 0.48 |
| | | To lower | 0.00 | 1.00 |
| | | To upper | 0.00 | 1.00 |
| | Knowledge | Country capital | 0.18 | 0.76 |
| | | Location continent | 0.01 | 0.72 |
| | | Location religion | 0.08 | 0.82 |

| Model | Task type | method
Task name | Baseline | Regular |
|---|---|---|---|---|
| | | Person language | 0.00 | 0.82 |
| | Linguistic | Antonyms | 0.37 | 0.76 |
| | | Plural singular | 0.13 | 0.78 |
| | | Present simple gerund | 0.00 | 0.96 |
| | | Present simple past simple | 0.03 | 0.92 |
| | Translation | En es | 0.10 | 0.76 |
| | | En fr | 0.16 | 0.60 |
| | | Es en | 0.08 | 0.82 |
| | | Fr en | 0.10 | 0.82 |
| Pythia 6.9B | Algorithmic | List first | 0.43 | 0.98 |
| | | List last | 0.08 | 0.98 |
| | | Next letter | 0.01 | 0.86 |
| | | Prev letter | 0.04 | 0.32 |
| | | To lower | 0.00 | 1.00 |
| | | To upper | 0.00 | 1.00 |
| | Knowledge | Country capital | 0.21 | 0.82 |
| | | Location continent | 0.01 | 0.78 |
| | | Location religion | 0.10 | 0.80 |
| | | Person language | 0.01 | 0.80 |
| | Linguistic | Antonyms | 0.33 | 0.74 |
| | | Plural singular | 0.14 | 0.88 |
| | | Present simple gerund | 0.00 | 0.94 |
| | | Present simple past simple | 0.02 | 0.96 |
| | Translation | En es | 0.11 | 0.70 |
| | | En fr | 0.21 | 0.60 |
| | | Es en | 0.06 | 0.82 |
| | | Fr en | 0.14 | 0.74 |
| RWKV 0.169B | Algorithmic | List first | 0.48 | 0.46 |
| | | List last | 0.10 | 0.19 |
| | | Next letter | 0.10 | 0.29 |
| | | Prev letter | 0.00 | 0.03 |
| | | To lower | 0.00 | 0.54 |
| | | To upper | 0.00 | 0.85 |
| | Knowledge | Country capital | 0.17 | 0.10 |
| | | Location continent | 0.02 | 0.67 |
| | | Location religion | 0.10 | 0.60 |
| | | Person language | 0.01 | 0.31 |
| | Linguistic | Antonyms | 0.10 | 0.08 |
| | | Plural singular | 0.12 | 0.24 |
| | | Present simple gerund | 0.00 | 0.25 |
| | | Present simple past simple | 0.00 | 0.20 |
| | Translation | En es | 0.04 | 0.18 |
| | | En fr | 0.11 | 0.24 |
| | | Es en | 0.08 | 0.10 |
| | | Fr en | 0.17 | 0.10 |
| RWKV 0.43B | Algorithmic | List first | 0.40 | 0.73 |
| | | List last | 0.17 | 0.48 |
| | | Next letter | 0.09 | 0.66 |
| | | Prev letter | 0.00 | 0.01 |
| | | To lower | 0.00 | 1.00 |

| Model | Task type | method
Task name | Baseline | Regular |
|---|---|---|---|---|
| | | To upper | 0.00 | 1.00 |
| | Knowledge | Country capital | 0.10 | 0.37 |
| | | Location continent | 0.00 | 0.56 |
| | | Location religion | 0.07 | 0.71 |
| | | Person language | 0.01 | 0.64 |
| | Linguistic | Antonyms | 0.16 | 0.56 |
| | | Plural singular | 0.10 | 0.22 |
| | | Present simple gerund | 0.00 | 0.51 |
| | | Present simple past simple | 0.01 | 0.64 |
| | Translation | En es | 0.04 | 0.41 |
| | | En fr | 0.18 | 0.39 |
| | | Es en | 0.08 | 0.57 |
| | | Fr en | 0.08 | 0.49 |
| RWKV 1.5B | Algorithmic | List first | 0.51 | 0.97 |
| | | List last | 0.26 | 0.78 |
| | | Next letter | 0.20 | 0.81 |
| | | Prev letter | 0.00 | 0.05 |
| | | To lower | 0.00 | 1.00 |
| | | To upper | 0.00 | 1.00 |
| | Knowledge | Country capital | 0.13 | 0.41 |
| | | Location continent | 0.02 | 0.58 |
| | | Location religion | 0.04 | 0.73 |
| | | Person language | 0.00 | 0.79 |
| | Linguistic | Antonyms | 0.37 | 0.70 |
| | | Plural singular | 0.03 | 0.69 |
| | | Present simple gerund | 0.00 | 0.71 |
| | | Present simple past simple | 0.00 | 0.74 |
| | Translation | En es | 0.11 | 0.60 |
| | | En fr | 0.15 | 0.61 |
| | | Es en | 0.13 | 0.82 |
| | | Fr en | 0.14 | 0.73 |
| RWKV 3B | Algorithmic | List first | 0.29 | 0.96 |
| | | List last | 0.43 | 0.88 |
| | | Next letter | 0.02 | 0.56 |
| | | Prev letter | 0.02 | 0.08 |
| | | To lower | 0.00 | 1.00 |
| | | To upper | 0.00 | 1.00 |
| | Knowledge | Country capital | 0.08 | 0.79 |
| | | Location continent | 0.00 | 0.74 |
| | | Location religion | 0.02 | 0.74 |
| | | Person language | 0.00 | 0.82 |
| | Linguistic | Antonyms | 0.36 | 0.74 |
| | | Plural singular | 0.09 | 0.82 |
| | | Present simple gerund | 0.00 | 0.83 |
| | | Present simple past simple | 0.01 | 0.83 |
| | Translation | En es | 0.07 | 0.61 |
| | | En fr | 0.13 | 0.63 |
| | | Es en | 0.01 | 0.84 |
| | | Fr en | 0.05 | 0.75 |

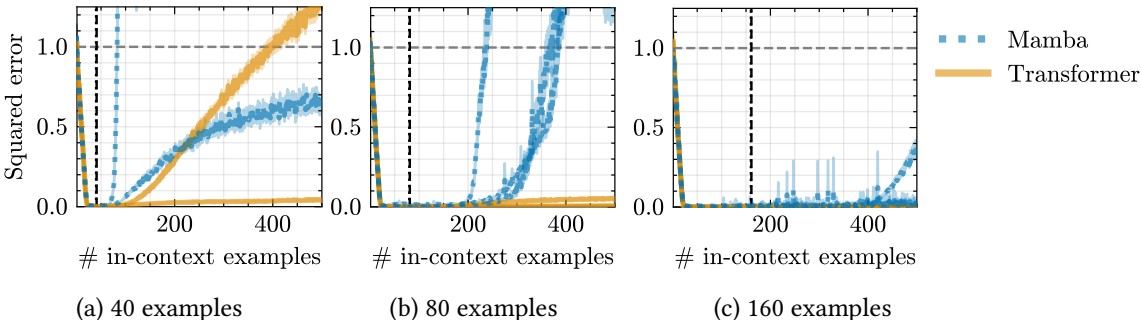

(a) 40 examples        (b) 80 examples        (c) 160 examples

Figure 10: Length extrapolation performance for varying number of in-context examples used for testing and training (dashed vertical lines) on the skewed linear regression task. We report 3 training runs for each plot and model

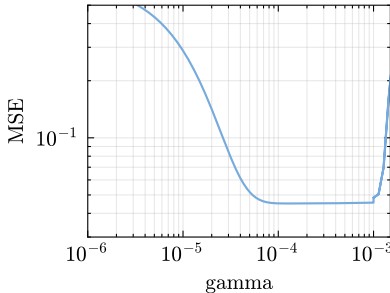

Figure 11: Results for grid search over $\gamma$ for GD++. We optimized $\gamma$ in order to have optimal average performance across tasks of GD++ at iteration 24 (the same number of iterations reported in Figure 3d). We picked a value of approximately $2 \times 10^{-4}$

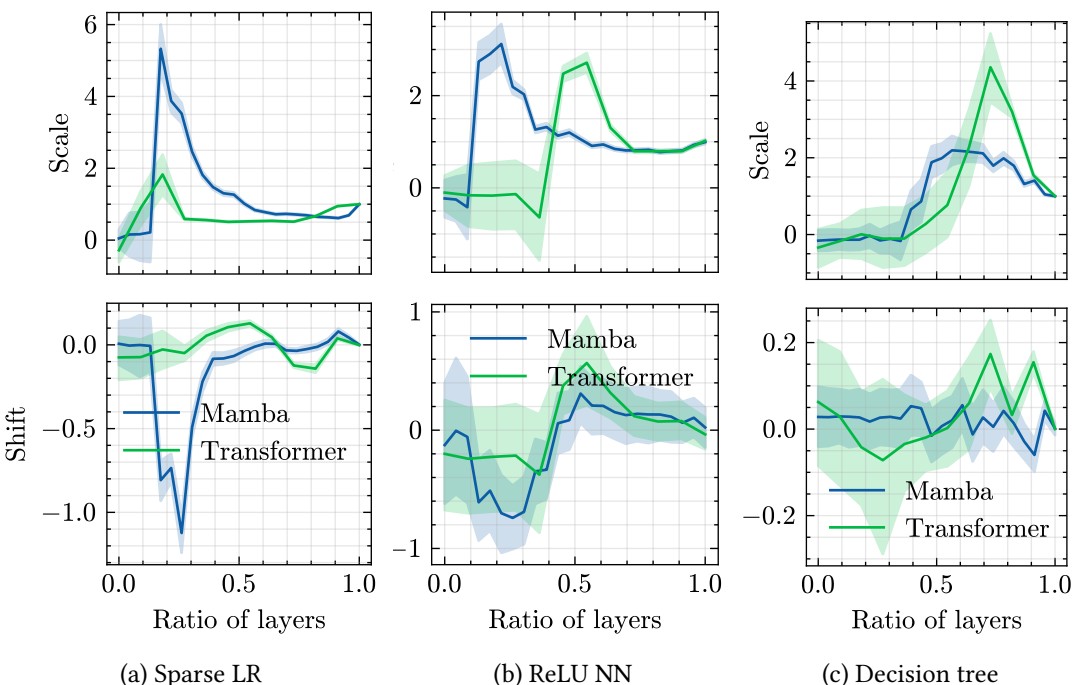

(a) Sparse LR  (b) ReLU NN  (c) Decision tree

Figure 12: Estimated scale and shift by the linear probing model used in Section 3.2.

