# OpenReview forum: "Is Mamba Capable of In-Context Learning?"
_automl.cc/AutoML/2024/Conference — AutoML 2024_

### Official Review · Reviewer_h7nq · 2024-03-22

**Potential Impact On The Field Of Automl Rating:** 3
**Technical Quality And Correctness Rating:** 3
**Clarity:** This paper is well-written and organi…
**Clarity Rating:** 3
**Actions Required To Increase Overall Recommendation:** see above.

**Summary Of Contributions:**

This paper investigates the In-Context Learning (ICL) capabilities of the Mamba architecture, a state space model that scales better than transformers with respect to input sequence length.

**Overall Review:**

Paper Strength
1. This paper provides empirical evidence of Mamba's comparable performance to transformer models in ICL tasks.
2. It offers a novel analysis of Mamba's mechanism for solving ICL problems through probing studies.
3. The research opens up possibilities for generalizing in-context learned AutoML algorithms to long input sequences.

Paper Weakness
Unclear Impact on AutoML: The paper suggests that Mamba could be used for generalizing in-context learned AutoML algorithms to long input sequences. However, the relevance of the paper's focus to the AutoML community is not clear to me.

**Potential Impact On The Field Of Automl:**

This paper investigates the In-Context Learning (ICL) capabilities of the Mamba architecture, a state space model that scales better than transformers with respect to input sequence length. The relevance of the paper's focus to the AutoML community is not clear to me.

**Review Confidence:**

3

**Review Rating:**

7

**Review Summary:**

This paper presents interesting findings on Mamba's capabilities for ICL tasks. The authors might consider addressing the limitations above.

**Technical Quality And Correctness:**

This paper is generally technically sound, with a clear methodology and thorough experimental evaluation. The comparison with transformer models and the probing analysis provide a solid basis for the claims made.

---

### Official Review · Reviewer_fG3t · 2024-03-27

**Potential Impact On The Field Of Automl Rating:** 3
**Technical Quality And Correctness Rating:** 2
**Clarity Rating:** 2

**Summary Of Contributions:**

&nbsp;

The authors investigate the question as to whether Mamba is a competitive approach to transformers for in-context learning (ICL). The authors evaluate Mamba's ICL capabilities on function approximation and NLP tasks. Additionally, the authors provide diagnostic experiments to examine the mechanism of ICL on a layer-by-layer basis in both Mamba and transformers.

&nbsp;

**Details Of Ethical Concerns:**

&nbsp;

No concerns.

&nbsp;

**Actions Required To Increase Overall Recommendation:**

&nbsp;

I would like to see a clearer presentation and exposition of the experiments. As things currently stand, there is some missing context and it is challenging to evaluate the findings. It would also be nice to see experiments directly showcasing applications of ICL using Mamba.

&nbsp;

**Clarity:**

&nbsp;

1. In the abstract it is stated that, "The surprising generalization capabilities of foundation models have been enabled by in-context learning (ICL)". This statement might be misconstrued to mean that ICL is SOLELY responsible for the generalization capabilities of foundation models and may warrant a revision.

2. Line 41, typo, "a recurrent mode".

3. Line 78, typo in the definition of \bar{C}.

4. It is worth stating explicitly in Section 2 that A, B, and C are matrices. It is also worth specifying their dimensionality at this point. Albeit this is only introduced later in the original Mamba paper, it would aid in clarity to fully define A, B, and C at the point they are introduced. In the current presentation for instance, it is very difficult for the reader to determine that the D parameter is a scalar as seems to be the case from the appendix?

5. Line 124, linear functions in the set F.

6. Line 128, is there a missing summation in the definition of the MSE?

7. In Figures 1 and 2, the presentation could be improved if the size of the y-axis label was the same between figures.

8. Line 146, broken reference.

9. Line 184, typo in "sparse".

10. Line 265, typo in "linear".

&nbsp;

**Overall Review:**

&nbsp;

While shedding light on the mechanism of ICL in Mamba is interesting, the link to AutoML is tenuous and it would be nice to see explicit experiments where the findings are directly used in an AutoML context.

&nbsp;

**Potential Impact On The Field Of Automl:**

&nbsp;

The link to AutoML is tenuous. The authors propose that the findings may be relevant for the AutoML domain via transformer-based models such as Optformer albeit no experiments are performed explicitly for AutoML.

&nbsp;

**Reproducibility:**

&nbsp;

It would be beneficial for users of the codebase if the code was more heavily documented using for example the PEP 257 guidelines.

&nbsp;

**Review Confidence:**

4

**Review Rating:**

6

**Review Summary:**

&nbsp;

Given the lack of clarity regarding the experimental setup, I would encourage the authors to revise the exposition of the paper and potentially add experiments in an AutoML setting to make the work relevant for the conference. I would be open to hearing the authors' opinions on where the current work might be a good fit (without a connection to AutoML) with reference to the call for papers.

Update: Now able to edit score, have upgraded!

&nbsp;

**Technical Quality And Correctness:**

&nbsp;

1. In Section 3, why is a Gaussian or skewed Gaussian distribution used to sample the inputs in place of a uniform distribution?

2. The experimental section lacks many details necessary for the reader to interpret the results from the submission alone. What is the ratio of layers in Figures 3 and 4? What are random quadrants and half subspace (random quadrants are described in the appendix)? Why does there appear to be a worsening of performance in all methods (including linear regression) as the number of in-context examples is increased? Although this is discussed in Section 3.2 it would be nice to understand this phenomenon in further detail.

&nbsp;

---

### Official Review · Reviewer_FWoA · 2024-03-28

**Potential Impact On The Field Of Automl Rating:** 3
**Technical Quality And Correctness:** The approach and experiments are soun…
**Technical Quality And Correctness Rating:** 4
**Clarity Rating:** 3

**Summary Of Contributions:**

The paper explores Mamba's in-context learning (ICL) capabilities. The authors demonstrate that Mamba is capable of ICL and outperforms other state space models (SSMs) while performing similarly to Transformers. The evaluation is conducted using simple function classes and NLP tasks.

**Actions Required To Increase Overall Recommendation:**

To confirm their findings, the authors can conduct additional experiments with more complex tasks. They can also use the Appendix to improve the related work section. Background on ICL capabilities in transformer-based models can benefit readers.

**Clarity:**

The paper is well-written. However, the Related Work section can be improved. For instance, it can be moved to the Appendix to discuss better existing research, e.g., ICL capabilities in transformer-based models.

**Overall Review:**

Strengths:  The paper provides empirical evidence and demonstrates Mamba's ICL capabilities with simple tasks. The methodology and analysis are sound.
Weaknesses: The experiments on simple tasks might raise concerns about the claims' strength.  The related work section has room for improvement. ICL capabilities in transformer-based models can be discussed in the Appendix.

**Potential Impact On The Field Of Automl:**

The paper is important for the AutoML research community because its insights can motivate improvements in existing solutions or the development of new AutoML solutions that benefit from Mamba's ICL capabilities.

**Review Confidence:**

5

**Review Rating:**

7

**Review Summary:**

The paper reasonably investigates Mamba, a popular alternative architecture to Transformers. The focus is on its in-context learning (ICL) capabilities, and the authors do a good job of exploring Mamba's ICL strengths and limitations. Although I have some concerns regarding the complexity of the tasks used for their experiments, the paper presents a good initial investigation of Mamba's ICL capabilities, and I recommend a weak acceptance.

---

### Official Review · Reviewer_1EJB · 2024-03-29

**Potential Impact On The Field Of Automl:** 1. Mamba's demonstrated ability to pe…
**Potential Impact On The Field Of Automl Rating:** 3
**Technical Quality And Correctness:** 1. The study's focus on specific type…
**Technical Quality And Correctness Rating:** 2
**Clarity Rating:** 3
**Actions Required To Increase Overall Recommendation:** Please address things raised in the w…

**Summary Of Contributions:**

The paper "Is Mamba Capable of In-Context Learning?" explores the in-context learning (ICL) capabilities of the Mamba architecture, a state space model proposed as an alternative to transformer models for processing long input sequences. The study investigates Mamba's performance across tasks involving simple function approximation and complex natural language processing (NLP) challenges, comparing its ICL abilities to those of transformer models, its predecessor S4, and a recurrent neural network architecture, RWKV. The results demonstrate that Mamba matches or even outperforms these models in ICL tasks, particularly in handling long sequences efficiently. The analysis suggests that Mamba, like transformers, employs an iterative optimization process to refine its internal representations for task solving.

**Clarity:**

The work has demonstrated clarity in explaining the problem identification, experimental setup as well as results and conclusions.

**Overall Review:**

It would be valuable to look into recent advancements and discussions around in-context learning capabilities in large language models, state space models versus transformers for long sequence handling, and novel methodologies for analyzing model behaviors. This additional information could offer alternative perspectives on Mamba's strengths and limitations, as well as potential directions for future work in enhancing its performance and understanding its mechanisms of action.

**Reproducibility:**

Reproducible

**Review Confidence:**

3

**Review Rating:**

8

**Review Summary:**

The authors investigate Mamba, a popular alternative architecture to Transformers with a focus on In-Context Learning.The experiments on simple tasks raises questions about comprehensiveness of the study.
I am a little unsure if this paper has direct relevance to AutoML

---

### Meta-Review · Area_Chair_xZWi · 2024-04-23

**Paper Recommendation:** Accept
**Confidence:** 4

**Metareview:**

This paper investigates the in-context learning capabilities of Mamba. Experiments demonstrate that Mamba is able to successfully perform in-context learning and shed light on the optimization process by which it does so. Reviewers agreed that the paper is well-written and the experimental methodology is sound. Some doubt was expressed about the extent to which the results are relevant to AutoML. However, the authors clarified the connections to meta-learning and potential applications of the work towards improving AutoML algorithms such as Optformer and TabPFN. I therefore recommend acceptance.

---

### Decision · Program_Chairs · 2024-04-29

**Decision:**

Accept

**Comment:**

Thank you for submitting your paper. We are happy to tell you that we accept your paper to the main track. See you in Paris.